# What factors have influenced quality of life in people with dementia and their family carers during the COVID-19 pandemic: a qualitative study

Stephanie Daley ![ORCID],[1] Nazire Akarsu,[2] Elise Armsby,[2] Nicolas Farina,[1] Yvonne Feeney,[1] Bethany Fine,[2] Laura Hughes ![ORCID],[1] Joanna Pooley,[2] Naji Tabet,[1] Georgia Towson,[2] Sube Banerjee[1,3]

¹Centre for Dementia Studies, Brighton and Sussex Medical School, Brighton, UK
²Sussex Partnership NHS Foundation Trust, Worthing, UK
³Faculty of Health, University of Plymouth, Plymouth, UK

**Correspondence to**
Dr Stephanie Daley;
s.daley@bsms.ac.uk

## ABSTRACT

**Objectives** The COVID-19 pandemic has led to significant disruption to health and social care services. For people with dementia and their family carers this is problematic, as a group who rely on timely and responsive services to live well with the condition. This study has sought to understand how COVID-19 has affected the quality of life of people diagnosed with dementia and their family carers.

**Design** Our mixed-methods study was nested in a larger cohort study of an education programme, Time for Dementia.

**Setting** The study took place in the South-East of England.

**Participants** Existing study participants, family carers were approached about the COVID-19 nested study. A purposeful sample of participants were invited to take part in in-depth qualitative interview. The sample included family carers in a range of different caring situations.

**Measurement** Interviews were undertaken remotely by telephone. Interviews sought to understand quality of life before the pandemic, impact of the restrictions on both the person with dementia and family carer, role of services and other agencies as well as supportive factors. Data were analysed using thematic analysis.

**Results** 16 family carers were interviewed. Seven themes were identified from our analysis: (1) decreased social interaction; (2) reduced support; (3) deteriorating cognitive and physical health for the person with dementia; (4) decreased carer well-being; (5) difficulties understanding COVID-19 restrictions; (6) limited impact for some and (7) trust and relationship with care home. There was little change between themes during the first and second wave of national lockdowns.

**Conclusions** Our study provides an understanding the short-term impact of COVID-19 on the quality of life of people with dementia and their family carers. Our findings suggest that recovery between the first and second wave of the restrictions did not automatically take place.

## INTRODUCTION

The COVID-19 pandemic and healthcare policy has been hugely impactful on people with dementia. Dementia has been shown to contribute towards a higher risk of mortality from COVID-19,[1–3] and healthcare policy towards those with dementia, particularly to those living in care homes has been shown to contribute towards increased mortality, distress and poor care delivery.[4 5] Additionally, regional and national lockdowns have led to significant disruption to health and social care services, which in some area still remain. For people with dementia and their family carers this is problematic. There are currently 850 000 people with dementia living in the UK[6] who rely on timely and responsive services to live well with the condition. The abrupt closure and reduction of many health and community support services arising from the pandemic is of concern for this vulnerable group. Many governments introduced a range of restrictions (shielding for the most medically vulnerable, self-isolation and social distancing) to help reduce the spread of COVID-19, and to protect older and vulnerable people. Commentaries exist about how the these restrictions may be affecting people with dementia and carers,[7] however, the full

and longer-term impact of these restrictions has yet to be assessed.

## Strengths and limitations of this study

► Strengths of this study are the inclusion of a range of family carers across different caring situations, as well as data collection in 'real time'.
► A further strength is that the study did not have an a priori hypothesis, and as such took an inductive 'bottom up' approach to data collection and analysis, with strong lived experience involvement.
► Limitations include the use of proxy reports from family carers about the impact of the pandemic on the quality of life of people with dementia, rather than self-report
► The study sample lacks ethnic diversity as well as those with a recent diagnosis of dementia.

Emerging research highlights that along with the disruption to formal dementia services, restrictions have led to a further loss of support from family and friends for people with dementia and family carers.[8] Since the first national lockdown in March 2020 in the UK, there has been an increased number of people telephoning the support services provided by the Alzheimer's Society due to concerns about increased cognitive and functional deterioration in family members with dementia.[9] Additionally, the UK advocacy group; 3 Nations Dementia Working Group) reported that that people with dementia attending their regular webinars expressed significant concern about the withdrawal of services leading to a sense of abandonment, along with anxiety about the potential for a permanent loss of services.[10] This theme of abandonment was also identified in a review of over 5000 Twitter tweets by people with dementia and their carers during the pandemic, due in part to the enforced isolation and lack of contact with others, particularly for those in care homes.[11]

Empirical data are emerging. One qualitative study which looked at the impact of reduced social support with 42 carers and 8 people with dementia identified that it was the 'suddenness' of the reduction in services that had significantly affected both family carers and people with dementia.[12] This research identified three key themes: loss of control, uncertainty (about current and future provision), and a need to adapt to 'the new normal' (including remote rather than face to face contact). A survey of 191 family carers of people with dementia found an 83% increase of caring tasks as a consequence of COVID-19, with over 50% reporting that they felt unable to fulfil the caring role adequately.[13] Increased carer stress, fatigue and burn-out has also been found in other studies.[3 14 15] In one study 21 family carers, severe loneliness, and an overwhelming sense of loss and concern about worsening cognition in the person with dementia was identified.[16] Concern about cognition and the impact of restrictions on cognition and neuropsychiatric symptoms has been identified more widely.[17 18] For example, one spanish study identified an increase in neuropsychiatric symptoms in people with Alzheimer's Disease and Mild-Cognitive impairment during 5 weeks of lockdown, with agitation, apathy and aberrant motor behaviour being most prominent.[19]

The emerging literature highlights the substantial impact of the loss of both formal and voluntary sector services, a fear of services not resuming, increased carer burden, and concern about increased cognitive decline in the person with dementia. We have been keen to understand the specific impact of the pandemic on quality of life (QoL) given that living with a diagnosis of dementia, or caring for somebody with dementia is highly impactful on all of the factors known to affect QoL, such as physical health, psychological state, independence, social relationships, personal beliefs and environmental

supports.[20] While factors influencing QoL in both people with dementia and their carers differ, their outcomes are inextricably linked; with carers being a vital determinant of QoL and positive outcomes for people with dementia, and vice versa. A recent scoping review sought to assess the impact of the pandemic on QoL of people with dementia and their carers[21] through the assessment of both care provision and QoL data for both groups. This review identified three areas in which the pandemic had impacted on QoL; unmet and increasing care needs, increased stress and burden in carers of people with dementia, and finally a range of demographic factors, such as disease severity. This is perhaps not surprising, however, we were keen to understand the impact more broadly and the relationship between factors. Therefore, in this qualitative study, we have aimed to understand how COVID-19 has affected the QoL of people diagnosed with dementia and their family carers

## METHODS

### Study design

Our study was nested in a larger cohort study, Time for Dementia.[22] Time for Dementia is an undergraduate education programme in the South-East of England, whereby people with dementia and their carer teach healthcare students about living with the condition. During the pandemic period, there has been remote contact (videoconferencing or telephone) between carers, the person with dementia and their family carer on a termly basis.

### Sample and setting

Study participants lived in Kent, Surrey and Sussex. The region has the highest proportion of older people in the UK, with 50000 people current living with dementia. The region is an area of extremes; it includes both rural areas that tend to be more affluent but more socially isolated, and urban areas that have higher levels of deprivation. Notably, the region includes coastal areas that feature in the highest decile of deprivation.

We recruited family carers of people with a diagnosis of dementia from the Time for Dementia study. The study definition of family carers included married and unmarried partners, children, siblings, extended family and close friends. The term 'family carers' is since it was preferred by our lived experience advisory group (LEAP). We sampled for variation in characteristics of family carers, using the socio-demographic information provided by participants. From discussion with our LEAP, we agreed that these characteristics should include: resident and non-resident family carers, spousal and non-spousal family carers, those who were shielding, and family carers of people with dementia in care homes. Due to the difficulty in assessing capacity over the telephone, people with dementia were not interviewed as part of this study.

## Procedure

Participants in the Time for Dementia evaluation were approached by a research worker about the COVID-19 nested study. Those who were interested were sent a study information sheet, and verbal consent was obtained by those who wished to take part. Quantitative measures of QoL were undertaken, along with sociodemographic information and are reported separately. A purposeful sample of participants were asked about undertaking an in-depth qualitative interview. The sample included family carers in a range of different caring situations.

Topic guides were developed from an initial review of the literature, and key topic areas included: life before the pandemic, impact of the restrictions on both the person with dementia and family carer, role of services and other agencies and supportive factors (see online supplemental file 1). Additionally, the topic guide for interviews were amended to explore change over the pandemic period, and also for family carers of people living in a care home.

Interviews were carried out between May and November 2020 by JP, SD and NA. Twelve interviews were undertaken in the first lockdown (May and June 2020), and 10 interviews were undertaken during the second lockdown (November 2020). Six participants were interviewed twice, and family carers of people living with dementia living in a care home (n=4) were interviewed once during the second lockdown. Interviews were stopped when data reached thematic saturation. Interviews were undertaken by telephone and lasted between 30 and 60 min. All interviews were audiorecorded, transcribed verbatim and checked for accuracy.

## Analysis

Data analysis commenced in July 2020 using thematic analysis. The process started with descriptive coding of two transcripts by two researchers (SD and JP) who manually coded the transcripts by giving descriptive codes to meaningful segments of text. The researchers met to review their respective preliminary codes and identify areas of differences and to develop an initial framework. The remaining 10 transcripts (from the first round of interviews) were coded to develop a focused framework. The topic guide was amended at this stage, to gain further understanding of developing themes and to explore change over time. During the second phase of the analysis (second round of interviews), six transcripts were coded by SD and JP. GT and SD undertook a separate analysis of the care home transcripts, using the existing study coding framework, generating additional codes and themes as necessary.

The computer software package, NVivo V.12 (QSR International, 2018) was used to enable the systematic collation and review of the data grouped within each code. The researchers met regularly to compare coding and data between existing and new transcripts in order to check on the use of codes for consistency and areas of uncertainty. The researchers also jointly reviewed data collated within each code, to identify relationships between codes, as well as areas of uncertainty, coming to an agreement once discussed. In the last phase of the analysis, the care home data was incorporated, and a final framework was produced from which seven over-arching themes were identified.

Rigour in the research process was supported in two ways. First, the lead researcher (SD) met frequently with the researchers for academic supervision to review coding, the development of themes and reflexivity. Second, all researchers kept fieldwork diaries.

## Patient and public involvement

People with dementia and their family carers were involved throughout the research process though a specific LEAP which was established for the study. The group reviewed the study design, participant-facing documentation, the topic guide on two occasions as well as emerging and final themes. The group have also been involved in developing dissemination material for participants, as well as for local services and stakeholders.

## RESULTS

The 16 participants were predominantly white British/European (94%), spousal carers (64%). Half of participants were coresident with the person with dementia (50%).

Table 1 shows the participant characteristics.

## Themes

Our analysis identified seven themes: (1) decreased social interaction, (2) reduced support, (3) deteriorating cognitive and physical health, (4) decreased carer well-being (5) difficulties understanding COVID-19 restrictions, (6) limited impact for some and (7) trust and relationship with care home. Themes are illustrated in more depth in box 1.

These themes were all identified in relation to perceived impact of COVID-19 on the QoL of the person with dementia and family carer. For the majority of themes, there was little change between the first and second lockdown. With the exception of family carers of people with dementia living in care homes, those interviewed highlighted day-to-day life before the pandemic as including opportunities for social interaction and for support for both the person with dementia and family carer with benefits to each. Many of the themes identified are interrelated both between themselves and between the person with dementia and their carer, as shown in figure 1.

## Theme 1: decreased social interaction

All family carers reported that the significant reduction in face to face social interaction and contact with others impacted on QoL for themselves and the person with dementia. Feelings of loneliness and isolation were frequently reported by family carers, especially for those who would have ordinarily relied on social contact as a source of respite from their caring role. Lack of social

**Table 1** Participant characteristics

| Characteristic | Mean (SD) (range) | n | % |
|---|---|---|---|
| Age | 65.63 (9.88) (48–83) | 16 | |
| Gender | | | |
| Female | | 9 | 56.3 |
| Ethnicity | | | |
| White British/European | | 15 | 93.8 |
| Asian/Asian British | | 1 | 6.3 |
| Marital status | | | |
| Married | | 10 | 62.5 |
| Cohabiting | | 2 | 12.5 |
| Separated/divorced | | 2 | 12.5 |
| Never married/single | | 2 | 12.5 |
| Relationship to person with dementia | | | |
| Spouse/partner | | 10 | 62.5 |
| Son/daughter | | 6 | 37.5 |
| Currently employed | | | |
| Yes | | 4 | 25.0 |
| Current/last occupation | | | |
| Managers/directors/officials | | 3 | 18.8 |
| Professional | | 3 | 18.8 |
| Associate professional/technical | | 2 | 12.5 |
| Administrative/secretarial | | 5 | 31.3 |
| Skilled trades | | 2 | 12.5 |
| Caring/leisure/service | | 1 | 6.3 |
| Geographical location | | | |
| Urban | | 1 | 6.3 |
| Rural | | 7 | 43.8 |
| Suburbs | | 8 | 50.0 |
| Person with dementia | | | |
| Type of dementia | | | |
| Alzheimer's disease | | 4 | 25.0 |
| Vascular dementia | | 3 | 18.8 |
| Early onset Alzheimer's disease | | 2 | 12.5 |
| Mixed | | 7 | 43.8 |
| Residence | | | |
| Coresident with carer | | 8 | 50.0 |
| Non-resident | | 8 | 50.0 |
| Person with dementia | | | |
| Community dwelling | | 12 | 75.0 |
| Care home resident | | 4 | 25.0 |
| Time since diagnosis | 3.4 years (2.20) (1.1–7.5) | | |
| Less than 1 year | | 0 | 0.0 |

Continued

**Table 1** Continued

| Characteristic | Mean (SD) (range) | n | % |
|---|---|---|---|
| Between 1 and 3 years | | 9 | 56.3 |
| Between 3 and 5 years | | 2 | 12.5 |
| Over 5 years | | 5 | 31.3 |

interaction led to sense of reduced informal support (from others) and was impactful across the majority of other themes identified in the analysis.

Family carers expressed a sense of time being short for the person with dementia, and precious time with family members being lost. Concern about the potential risks of catching the virus appeared to result in isolation for many family carers who reported that they had avoided social opportunities, declined support from services both for themselves and the person with dementia.

Family carers of those in care homes were all spousal carers, and all reported that being unable to see the person with dementia was very impactful on their QoL, leading to significant feelings of loneliness.

### Theme 2: reduced support

Family carers stated that pre-COVID informal face-to-face support provided by friends and family was highly valued, and positively impactful on their own QoL as well as the QoL of the person with dementia. Since the COVID-19 pandemic, this type of support had reduced for many, primarily due to social distancing regulations, including shielding. In the absence of face-to-face support, family carers recognised the benefit of remote contact with friends and family. An increased reliance on technology (such as phone and video calls) was seen as being problematic for both the family and for the person with dementia. Family carers reported that it is more difficult to offload fully about current caring challenges by phone or video calls, either because the person with dementia was present or because there was less available time. For those with dementia who were unfamiliar with technology such as video calls, family carers reported that using them was often confusing, and of limited value.

A sharp reduction of services such as day care, respite, community groups and non-urgent healthcare was widely reported, accompanied by concerns about the future resumption of support services. For a small number of those interviewed, formal respite services had continued or had resumed, and this was highly valued.

Family carers also reported that some professional support groups adapted to provide some services remotely, and while this was valued, there was dissatisfaction with the depth of connection and support possible through remote contact.

Due to COVID-19, the process of obtaining a care home place had become more difficult, with three family carers reporting that the transition into the care home had ultimately been made following an acute hospital admission.

## Box 1  Summary of themes

**Themes/Quotes**

**Theme 1: Decreased Social interaction:**

*'very very isolating really, both for me and mum really erm, coz she misses, she obviously misses seeing everyone'* Family carer 12, resident son, (baseline)

*'um well I miss it, miss it terribly because I'm quite gregarious and I do miss it, probably y'know somebody, not having people to kind of talk to and have a laugh about things and um just generally just to offload really, um y'know that makes a difference'* Family carer 2, resident spouse, female (follow-up)

*'the other main one really was has not being able to see his grandchildren. And because it's hard for anybody to know you know and I've sat with him while he's cried and gone "I don't know if I'll ever see them again," you know and that that breaks my heart you know that's really tough. And so those kinds of things are really challenging'* Family carer 6, non-resident daughter, (baseline)

**Theme 2: Reduced support:**

*'but you know, it's just um when, when will I be able to get a break, that's what I don't know'* Family carer 10, resident spouse, female (baseline)

*'definitely (day care) being open …because that at least you know I'd go round and see mum in any case but then I'd take her down to [(day care]) and then in 3 hours hours' time she would you know had company, something to eat um and just you know, somebody else seeing her, that 3 hours hours to me on that Thursday and a Tuesday meant a lot'* Family carer 7, non-resident, daughter (baseline)

*'(peer support group) have never reopened. It was on Zoom but X (person with dementia) couldn't partake in Zoom, he found that really difficult…. So, to try and understand and relate to lots of people in boxes, he just sat there and then he would look at me and say "how long have I got to sit here?" I said well you haven't got too, he couldn't understand that or partake in it so we stopped that so he's had nothing really. The Zooms stopped'* Family carer 10, resident spouse young onset, female (follow-up)

*'we've got friends and they would help out so the wife would come and uh look after X (person with dementia) and I go out with, with the husband. Um that stopped of course because um we can't even have someone come round to the house'* Family carer 3, resident spouse, male (follow upfollow-up)

**Theme 3: Deteriorating cognitive & physical health:**

*'the lack of stimulation I'm certain has had an impact on her ability to um join things up. Certainly her memory has got worse. … Um she just doesn't know what day it is um sometimes she doesn't know if it's morning or afternoon. And that's certainly has gotten worse since we've gone on lockdown'* Family carer 3, resident spouse, male (baseline)

*'she's sitting down a lot more, not getting out and about and moving so, so she's got more aches and pains'* Family carer 7, non-resident, daughter (baseline)

*'not having the sorta will do doing things sorta thing, he's sorta getting not actually lazy but just getting the exhausted sorta doing something sorta, his routine is just laying down or sitting down sorta thing but not things sorta moving sorta doing anything'* Family carer 8, non-resident son, (baseline)

*'no, no it was, he was in decline before that'* Family carer 16, care home spouse, female

**Theme 4: Decreased carer wellbeingwell-being:**

*'also during that time her need for help with personal care has increased, so I'm more and more tied down and there's no easy way out*

Continued

## Box 1  Continued

*of that… I have less and less time because I'm less and less able to leave X (person with dementia) for any length of time, either to go out for a quick walk or even to do things in the house'* Family carer 3, resident spouse, male (follow-up)

*'with the pandemic, obviously mum's carer, my helpers who used to call round of an evening obviously most of them then couldn't go round so it was myself going round practically every evening'* Family carer 7, non-resident, daughter (baseline)

*'I've been very weepy when we were in lockdown, I've not been coping nearly as well as normal because it's been so hard because of having no time for me, and no break. I think I'm just exhausted, exhausted and no time to myself'* Family carer 10, resident spouse young onset, female (baseline)

*'you know it's, he just kinda like suffocates me all the time. And I just get, get really frustrated. And I just, sometimes I just don't wanna be here, I wanna be anywhere else but here'* Family carer 2, resident spouse, female (baseline)

**Theme 5: Difficulties understanding COVID-19:**

*'at that time, he was just really really unhappy because he didn't understand what was going on and why he couldn't do things and why our sons couldn't come round and yeah, I mean why he couldn't see our little grandson, why he couldn't get his haircut etc, … his lack of understanding of why he can't go and get his haircut and why he can't go to a shop and buy something that he wants. You know, when I say we can't do that he says "Why? Why on earth not?" You know he can't understand that and explain, I explain time and time again but often he doesn't get it'* Family carer 10, resident spouse young onset, female (follow upfollow-up)

*'and therefore she doesn't quite understand all of it. And uh just keeps repeating and repeating that you know 'Why can't we?'* Family carer 4, resident spouse, male (baseline)

**Theme 6: Limited impact for some:**

*'He doesn't, he doesn't, he's got no anxiety so he doesn't worry about people not being around the house coz he doesn't remember'* Family carer 11, resident, daughter (baseline)

*'she's just basically quite unaware of what's happening'* Family carer 3, resident spouse, male (follow-up)

**Theme 7: Trust and relationship with care home:**

*'and she's being looked after and it's the best place for her so therefore I suppose it, it, it does me help you know, it makes me feel better knowing that she's, she's okay you know… the homes very good, they put me straight onto X (person with dementia) and they let me know how she's getting on'* Carer 13, care home spouse, male

*'and they've never, I asked, I asked every week, "can I see X (person with dementia), can I see X (person with dementia)" "oh we don't know where the (I-Pad) tablet is, it must be in the manager's office". Hey ho they have lost it, it's now, I keep asking for this tablet and no one knows where it is'* Carer 15, care home spouse, female

### Theme 3: deteriorating cognitive and physical health

Many of the family carers interviewed expressed concern about increased cognitive decline in the person with dementia, linking this to a lack of stimulation, from reduced social interaction with others (family, friends and dementia support services). Several of carers interviewed noted that the decline in cognition could also

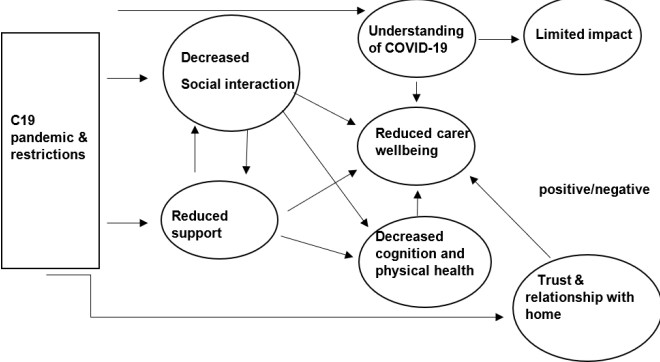

**Figure 1** Relationship between themes.

have been attributable due to the natural progression of the condition.

Where the person with dementia was community dwelling, family carers reported that reduced levels of social and support activity disrupted the routine of the person with dementia, negatively impacting on their motivation and fatigue levels.

Family carers reported worsening physical health in the person with dementia for those who were community dwelling. Reduced mobility was highlighted as a particular concern, linked to the wider lack of activity, along with an increased rate of joint pain, and for some, an increase in urinary tract infections.

### Theme 4: decreased carer well-being

For most family carers, the pandemic had led to increased caring tasks arising from increased dependence of the person with dementia and a reduced level (service or family/friend) support. Increasing caring, along with the lack of respite opportunities for a break had led to a sense of burnout for some. Feelings of frustration, anger and resentment from some of those interviewed was expressed, increasing over time between the first and second lockdown.

Additionally, for family carers, increased caring tasks and the need for direct supervision of the person with dementia impacted on their ability to meet their own needs. The majority of family carers interviewed described a loss of freedom in being able to undertake the activities which they wanted to undertake, and a sense of a loss of control and certainty. Some of those interviewed described feeling suffocated, lonely and frustrated.

### Theme 5: difficulties understanding COVID-19 restrictions

Difficulties in understanding the pandemic and its related restrictions was reported as being problematic for people with dementia. Carers reported that for some of those with dementia, the restrictions were often forgotten, or that they were too confusing, or difficult to understand. For many family carers, they reported that they frequently needed to explain the pandemic and remind the person with dementia about restrictions (eg; need to avoid close contact with others) when outside the home. This also applied to people with dementia in care homes, as

carers stated that they were constantly having to explain why they could not visit in person, or why they were not allowed to touch the person with dementia. This often led to the person with dementia becoming upset, or in some cases angry.

### Theme 6: limited impact from the pandemic

For a small number of people with dementia, carers reported that there was no perceived impact arising from the pandemic due their lack of understanding of the situation, and not remembering that they were not seeing family and friends less often.

### Theme 7: trust/relationship with care home

Where the person with dementia was in a care home, family carers identified the importance of maintaining a good relationship and trust with the care home as being pivotal to their QoL, due to the family carer being unable to visit the home. Carers stated that care homes who facilitated good communication through phone calls, video calls or emails enabled a greater sense of ease and confidence. Having confidence in the care provided, and knowing that the person with dementia was safe and well looked after was perceived as essential for family carer QoL. One participant described poor care being delivered to the person with dementia, along with poor communication as having a negative impact on their QoL.

Maintaining contact between the carer and the person with dementia was achieved in a variety of ways, including the use of garden 'pods' or telephone contact. However, one participant reported that no alternative means of maintaining contact with the person with dementia was provided. The ability to maintain contact with the person with dementia impacted carer QoL, and increased contact with the person with dementia was associated with enhanced satisfaction with care and improved QoL for carers.

### DISCUSSION

We have identified seven themes arising from the COVID-19 pandemic which family carers reported to have affected QoL for people with dementia and family carers.

Reduced social interaction was the most impactful factor influencing QoL, and while its identification as a factor is not surprising, its mediating role on other factors is of note. For people with dementia; the value of social relationships on QoL in terms of social support, social integration and companionship[23] is well known. For family carers, the positive impact of social interaction from family and social networks on QoL is also well established.[24–26] However, in terms of the impact of the COVID-19 pandemic, the effect appears to be two-pronged, first it is sudden yet ongoing sense of loss of social contact with meaningful others which is impactful, as well as actual impact of reduced contact with others. For the person with dementia, the reduced social contact

was seen by the majority of family carers as increasing cognitive impairment in the person with dementia, reinforcing the importance of social contact as a protective factor against cognitive decline.[27] Reduced social interaction with others, and the perceived impact of this on the cognition of the person with dementia increased the burden on family carers, along with the loss of personal freedom and a decreased ability to meet their own needs and their overall well-being. Loss of personal freedom has been identified as a factor in the QoL of family members of people with dementia.[24 26 28] Reduced family carer well-being is of concern, as if prolonged, this may impair the ability of family carers to look after the person with dementia, and may lead to an increased rate of transition into care homes. Our findings indicate that the resumption of face-to-face social support services for people with dementia and their family carers should be prioritised. Furthermore a lessening of restrictions will bring with it opportunities for families and friends to socialise, which of highly valued for this group The impact of reduced formal support on QoL is consistent with emerging research,[3 12 29 30] and fears about services not resuming.[10] Finally, difficulty understanding local guidance for restrictions, and for the person with dementia, the rationale behind the restrictions was also identified in an analysis of Twitter tweets[11]

Our finding that the social restrictions from the pandemic have negatively impacted on the physical health of the person with dementia has not yet been identified elsewhere, indicating that more research in this area is required. Interestingly, while worsening family carer physical health is factor known to influence the QoL of carers of people with dementia[31] this was not identified in this study, possibly because of the relatively short time period, or alternatively because it may have existed pre-COVID-19.

Within the context of a care home setting, the importance of trust and a good relationship between family carers and staff is clear. In our study, three of the four people with dementia in a care home had been transferred to the care home during the COVID-19 pandemic. Notwithstanding some of the wider challenges experienced by care homes in relation to COVID-19, the lack of face-to-face contact between family carers and staff can be seen as an additional obstacle to be overcome. The need for care homes to meaningfully involve family carers is care decisions is recognised as being significant in the development of both trust and a good relationship,[32] and ordinarily trust in care staff evolves over time as a process,[33] therefore, the absence of a 'normal' face-to-face relationship with family carers has been problematic. Further research into the experiences of care home staff and people with dementia living in care homes during the pandemic would be useful to understand these experiences more fully

Finally, while this study provides an understanding the short-term impact of COVID-19 on people with dementia and their family carers, we do not know what the longer-term impact might be. We also do not know how a return to 'normal' might be achieved, or what the 'new normal' might look like for people with dementia and their family carers, and what might help or hinder this process. It is also not clear from this study whether the impact of the pandemic is similar across different contexts, for example, with shorter or less restrictive periods of national lockdown or in different areas. It does appear that recovery from the impact of the first wave of restrictions did not automatically take place, and anecdotally we are aware that reductions in social support services has been prolonged, with many services still not having fully resumed. The provision of formal support during the pandemic to the study sample was highly variable. The increased use of technology and potential opportunities to deliver services to people with dementia and their carers during the pandemic is evident.[34–36] However, the efficacy of these new models of delivery is not known, and there remain very few data about how dementia services have responded overall to the pandemic, and how they might adapt going forward to meet the post-pandemic challenges for this group.

These findings have a number of implications for services for people with dementia and their carers during period of national lockdowns or enforced social restrictions. First, there is a need for services to proactively to identify people with dementia and their carers who are struggling, and from this to provide more targeted support. Second, services need to ensure that people with dementia remain connected to other people, and provide support where necessary to do this. For some people, this may include setting up or adapting communication technology. Third, care homes should maintain regular communication with carers, and should proactively support active connection between people with dementia and their carers. Fourth, services have a key role in developing and disseminating dementia friendly summaries of complex national or local guidance. Finally, services should encourage and enable people with dementia and their carers to keep physically active, thereby helping to maintain good physical health and mobility.

There are three strengths to this study. First, we were able to include a wide range of family carers in order to understand the impact of COVID-19 across a range of caring situations. Second, interviews were undertaken during both national lockdown periods with data being collected in 'real time', without reliance on recall. Third, the study did not have an a priori hypothesis, and as such took an inductive 'bottom up' approach to data collection and analysis, supported by strong lived experience involvement. This has helped to strengthen the validity of the findings and kept the study alert to the current concerns of those living with dementia during this unprecedented period.

There are four main limitations to this study. First, we have taken proxy reports from family carers about the impact of the pandemic on the QoL of people with dementia. QoL is a subjective appraisal and it is well known

that proxy reports assess QoL differently to self-appraisal by people with dementia.[37] It is therefore possible that people with dementia may have identified other factors as being more or less impactful during the pandemic. Proxy-reports for the people with dementia living in care homes are also less likely to be accurate, as contact between family carers and the person with dementia may have been too limited for a more informed assessment of QoL. Second, there is a lack of ethnic diversity in the study sample. The limits the generalisability of these findings to the wide group of black, Asian and other ethnic minority group members who have been disproportionately negatively affected by the pandemic overall, and for whom the impact for those living with dementia may be more substantial. Third, this study did not include family carers of somebody of a recent diagnosis of dementia. All of those interviewed had been caring for the person with dementia for a period of time and therefore, the factors identified were additional to their ongoing experience of living with dementia and the degree to the pandemic impacted on this. This study, therefore, does not specifically aid understanding to the impact of COVID-19 on the QoL of those with a recent diagnosis of dementia. Fourth, we did not collect demographic information on household income and/or deprivation therefore limiting the generalisability of the study findings to different context and reducing potential observations about influencing economic and social factors.

However, this study does provide a new and valuable information that contributes usefully to the emerging evidence base on the impact of the COVID-19 restrictions on people with dementia and their family carers. The finding and associated recommendations can help commissioners and service providers in developing and delivering the robust postpandemic support that is needed by this vulnerable group.

**Acknowledgements** We would like to thank the contribution of the members of the lived experience advisory group; Julia Fountain, Ellen Jones, Claire Parker, Fiona McGhee, Tina and Ian Wakeford. We would also like to thank the following staff for their contribution to the study; Dr Sam Robertson, Rachel Russell, Natalie Portwine, Tamsin Eperson, Alice Russell, Jacob Reichental, Anomita Karim, Marcela Carvajal

**Contributors** Conception and design: SD, SB and NT. Fata collection: NA, JP, BF, EA and GT. Analysis: SD, JP and GT: Manuscript drafting: SD, JP, BF, SB, NF, YF and NT. All authors read and approved the final manuscript. SD acts as a gaurantor forthe study.

**Funding** This study was funded by the National Institute for Health Research (NIHR) Applied Research Collaboration Kent, Surrey, Sussex (grant number: N/A). Funding was also provided by Sussex Partnership NHS Foundation Trust and Brighton and Sussex Medical School.

**Competing interests** None declared.

**Patient consent for publication** Not applicable.

**Ethics approval** This study involves human participants and was approved by Health Research Authority London Queen SquareID 15/LO/0046. Participants gave informed consent to participate in the study before taking part.

**Provenance and peer review** Not commissioned; externally peer reviewed.

**Data availability statement** Data are available on reasonable request. Data can be shared through direct contact with corresponding author.

**ORCID iDs**
Stephanie Daley http://orcid.org/0000-0002-5842-0492
Laura Hughes http://orcid.org/0000-0001-9530-7053

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
