## [Reviewer comments · BMJ Open]

ARTICLE DETAILS

TITLE (PROVISIONAL)	What factors have influenced quality of life in people with dementia and their family carers during the COVID-19 pandemic: a qualitative study
AUTHORS	Daley, Stephanie; Akarsu, Nazire; Armsby, Elise; Farina, Nicolas; Feeney, Yvonne; Fine, Bethany; Hughes, Laura; Pooley, Joanna; Tabet, Naji; Towson, Georgia; Banerjee, Sube

VERSION 1 – REVIEW

REVIEWER	Kurniawan, Andree Pelita Harapan University, Internal medicine
REVIEW RETURNED	17-Aug-2021

GENERAL COMMENTS	The authors have done qualitative study about dementia in covid 19 time. My comments are 1. In the background/introduction of this study should be noted that elderly patients especially with comorbidity related to ages was related to severity of COVID-19, for instances dementia and parkinson disease. DOI: 10.1007/s00406-020-01205-z ; doi: 10.1016/j.parkreldis.2021.04.0192. In the discussion should be added: regarding the the results: what were the suggestion or advices for the policy maker, health care workers to help dementia patients coping in this pandemic. Tele consultation? safe assist from the nurse or health care workers? family support?
--

REVIEWER	O'Connell, Megan University of Saskatchewan, Psychology
REVIEW RETURNED	18-Aug-2021

GENERAL COMMENTS	What factors have influenced quality of life in people with dementia and their family carers during the COVID-19 pandemic: a qualitative study is timely, well written, and I see no concerns regarding methodology (the analysis plan appeared quite rigorous). It is, nonetheless, a small sample which is a limitation. The literature review is very sparse – there are more empirical data on COVID and dementia (for example, I have 2 publications focussed on COVID and family carers for those with dementia that are several months old – and I have no doubt there are more published during the intervening period). I would suggest a more fulsome literature review that is not restricted to the term including quality of life – descriptive research of the impacts of COVID include the sphere quality of life and might not explicitly use this term. The discussion section is well done but might benefit from a brief discussion of the local context for these carers – different regions of the world experienced different lockdowns and social
---

	distancing requirements. Moreover, some of this varied based on the local COVID caseloads based on timing – I wonder how the authors would speculate their findings would apply to the different COVID contexts these carers found themselves living with.
--	--

VERSION 1 – AUTHOR RESPONSE

Reviewer 1	
In the background/introduction of this study should be noted that elderly patients especially with comorbidity related to ages was related to severity of COVID-19, for instances dementia and parkinson disease. DOI: 10.1007/s00406-020-01205-z ; doi: 10.1016/j.parkreldis.2021.04.019 2.	Thank you for this recommendation, we have added text about the wider impacts of Covid-19 on people with dementia on page 4 The COVID-19 pandemic and healthcare policy has been hugely impactful upon people with dementia. Dementia has been shown to contribute towards a higher risk of mortality from COVID-19 (1-3), and healthcare policy towards those with dementia, particularly to those living in care homes has been shown to contribute towards increased mortality, distress and poor care delivery
In the discussion should be added: regarding the results: what were the suggestion or advices for the policy maker, health care workers to help dementia patients coping in this pandemic. Tele consultation? safe assist from the nurse or health care workers? family support?	Thank you for this helpful suggestion, we have added a paragraph on recommendations for commissioners/clinicians in the discussion on page 14. These findings have a number of implications for services for people with dementia and their carers during period of national lockdowns or enforced social restrictions. First, there is a need for services to proactively to identify people with dementia and their carers who are struggling, and from this to provide more targeted support. Second, services need to ensure that people with dementia remain connected to other people, and provide support where necessary to do this. For some people, this may include setting up or adapting communication technology. Third, care homes should maintain regular communication with carers, and should proactively support active connection between people with dementia and their carers. Fourth, services have a key role in developing and disseminating dementia friendly summaries of complex national or local guidance. Finally, services should encourage and enable people with dementia and their carers to keep physically active, thereby helping to maintain good physical health and mobility.
Reviewer: 2	
The literature review is very sparse – there are more empirical data on COVID and dementia (for example, I have 2 publications focussed on COVID and family carers for those with dementia that are several months old – and I have no doubt there are more published during the intervening period). I would suggest a more fulsome literature review that is not restricted to the term including quality of life – descriptive research of the impacts of COVID include the sphere quality of life and might not explicitly use this term.	Thank you for this suggestion, we have enhanced the content of the literature review on pages 4 & 5 and also within the discussion on pages 13 & 14.

The discussion section is well done but might benefit from a brief discussion of the local context for these carers – different regions of the world experienced different lockdowns and social distancing requirements. Moreover, some of this varied based on the local COVID caseloads based on timing – I wonder how the authors would speculate their findings would apply to the different COVID contexts these carers found themselves living with.	This is a helpful suggestion. Information about the setting has been added in the methods section on page 6 as well as within the study limitations on page 13 & 15. Study participants lived in Kent, Surrey and Sussex. The region has the highest proportion of older people in the UK, with 50,000 people current living with dementia. . The region is an area of extremes; it includes both rural areas that tend to be more affluent but more socially isolated, and urban areas that have higher levels of deprivation. Notably, the region includes coastal areas that feature in the highest decile of deprivation. We did not collect demographic information on household income and/or deprivation therefore limiting the generalisability of the study findings to different contexts and reducing potential observations about influencing economic and social factors.
---	---

VERSION 2 – REVIEW

REVIEWER	O'Connell, Megan University of Saskatchewan, Psychology
REVIEW RETURNED	14-Dec-2021

GENERAL COMMENTS	The revised manuscript includes several improvements - a more fulsome and up to date literature review and a discussion of the context in terms of geography where the research was conducted. Moreover, the authors included a discussion of the limitations of their findings to other context- notably those with shorter lockdowns. The addition of the implications for services section in the discussion was helpful and put their findings into context. I will say, however, the recommendation that services should encourage and enable persons living with dementia and their carers to be physically active is important - but not a simple behaviour to enact - particularly in a pandemic during lockdowns. I recognize it is not the purpose of the paper, but commenting on indoor activities might make this point more pertinent to lockdowns. There are small typos (extra periods, missing periods, lack of capitalization of Spanish) that will likely be caught during typesetting. In sum, an interesting paper that describes the unique impacts of the pandemic on those living with dementia and their carers.
--